# Mapping Resilience in the Town Camps of Mparntwe

**Chris Tucker** [1,*] ⓘ, **Michael Klerck** [2] ⓘ **and Anna Flouris** [2]

1   School of Architecture and Built Environment, University of Newcastle, Callaghan 2308, Australia
2   Tangentyere Council, Alice Springs 0870, Australia; michael.klerck@tangentyere.org.au (M.K.); anna.flouris@tangentyere.org.au (A.F.)
*   Correspondence: chris.tucker@newcastle.edu.au

**Abstract:** From the perspective of urban planning, the history of the Town Camps of Mparntwe (Alice Springs) has made them a unique form of urban development within Australia; they embody at once a First Nation form of urbanism and Country, colonial policies of inequity and dispossession, and a disparate public and community infrastructure that reflects the inadequate and ever-changing funding landscape it has been open to. While these issues continue, this paper discusses the resilience of these communities through the Local Decision Making agreement, signed in 2019 between the Northern Territory Government and Tangentyere Council. One thing that has been critical to translating and communicating local decisions for government funding has been the establishment of an inclusive and robust process of participatory mapping—*Mapping Local Decisions*—where both the deficiencies and potential of community infrastructure within each Town Camp is being identified. As local community knowledge is embedded within these practices, so too are issues of health, accessibility, safety and a changing climate similarly embedded within the architectural and infrastructure projects developed for government funding. Being conceived and supported by local communities, projects are finding better ways to secure this funding, building on a resilience these communities have for the places they live.

**Keywords:** Town Camps; First Nation communities; topological mapping; community infrastructure; PPGIS; minimalism; architectural design

## 1. Introduction

We locally identify the process used to discuss, map and design community infrastructure within the Town Camp as *Mapping Local Decisions*. This method builds on previous qualitative research using public participation geographic information systems (PPGIS) that aims to utilise the rich text and synergies of dialogue [1] within communities. These place-based methodologies are uniquely positioned for this research [2]. English is often not a first language for the First Nation people who live in the Town Camps, with twelve local Aboriginal languages spoken in them. Meanwhile, drawing the landscape and telling its story is also an important cultural way to tell others about Country. Methods that graphically record the usual ways that individuals talk to each other in Town Camp communities maintain a vital feedback loop, ensuring that proposals are visually located in the places that they will directly affect and use symbols that can be clearly understood no matter an individual's literacy background.

The need for local decisions concerning community infrastructure to be robustly recorded also relates to the historically complex tenure of land within the 16 Town Camps of Alice Springs, which has allowed its community infrastructure to develop at a lower standard to that provided in the rest of Mparntwe. Subsequently, many residents live in unacceptable and unsafe conditions, with restricted access to adequate housing, health, education and employment opportunities [3] (pp. 116–119). The responsibility for funding this infrastructure and its maintenance reinforces an inequality that continues a history of exclusion that residents have fought against since the 1880s, when the First Nation

people of Central Australia began to be dispossessed of their traditional lands. Since that time, the residents of Town Camps have resisted a colonialism that sought their removal and assimilation [4] (p. 19). In the 1970's, Town Camps began to actively assert their rights, forming a council in 1977—Tangentyere Council Aboriginal Corporation—where a movement for independence, control and self-determination began and continues to this day [5] (p. xii). Most of the work presently undertaken by Tangentyere Council is aligned with action on the social, environmental and behavioural determinants of health and wellbeing, delivering programs throughout Central Australia. Through this, *Mapping Local Decisions* has been made possible through the strong relationship Tangentyere Council has with the communities of the Town Camps, and their cultural awareness in being able to engage with the communities in constructive discussion.

## 2. Local Decision Making

The Local Decision Making (LDM) agreement between Tangentyere Council and the NTG, initiated in 2019 and signed in 2020 [6], prioritized self-determination and community control within Town Camps [7]. The agreement includes objectives to respect "*the long established and strong systems of Town Camp governance and leadership in the Alice Springs Town Camps...to document the commitment by the NT Government and TCAC to work together to implement LDM in the Alice Springs Town Camps...and to identify the services and priorities over which Town Campers wish to have control and for which they wish to have responsibility*" [6] (p. 2). This agreement was a significant shift in the processes used for how decisions about the provision of community infrastructure were to be made. It followed on from the Federal government's Northern Territory National Emergency Response, a 2009 joint funding program between the Australian Federal Government and the Northern Territory Government (NTG) called the Strategic Indigenous Housing and Infrastructure Program (SIHIP) [8]. SIHIP was a government-led and -controlled initiative established to design and construct community infrastructure in a range of indigenous communities in the Northern Territory. The Town Camps were included in this funding; however, most expenditure related to the pressing need for housing (133 of 199 Town Camp houses were upgraded) [9], with very little funding for community infrastructure [10] (p. 3). Some improvements to the road and community infrastructure of the larger Town Camps— Yarrenyty Arltere, Ewyenper Atwatye and Nyewente—were made; however, none of the work completed met the standards outlined in the Alice Springs Town Council Subdivision Guidelines [11]. As the Tangentyere Council response to the 2016 Inquiry into Housing Repairs and Maintenance on Town Camps noted, this failure "*...is supported by the fact that the Alice Springs Town Council is unprepared to deliver Municipal and Essential Services on any Town Camp.*" [10] (p. 3).

Complicating the responsibility for community infrastructure is that the NTG identifies the land occupied by the Town Camps as 'Community Living' intended for " . . . *temporary and permanent accommodation, and non-residential facilities for the social, cultural and recreational needs of residents*" [12] (p. 27). While this zoning appears to cover most activities within Town Camps, the current leasing agreements do not allow for them to be maintained by local government in similar ways to other residential areas of Mparntwe. The 'Community Living' zoning also prevents economic development activities, which limits the potential and self-determinism of Town Camps. The recent *Town Camps Reform Framework* recognises this limitation and appears to have the aspiration to reform this, allowing land owners " . . . *to use and develop the land in line with community and resident aspirations*" [13] (p. 10). Public space in the Town Camp is highly valued, and for many residents the space outside of the house is treated as a *living room*; this recognises a relationship to Country, an externally-oriented lifestyle, and a requirement to accommodate long- and short-stay visitors [14]. The *wellness* of residents has also become an important consideration in Town Camps [15], and while issues of housing have a significant impact on wellness [16], a study to investigate the environmental determinants to health and wellbeing in remote communities by Chakraborty et al. [17] identified that aboriginal people living in remote

areas of the Northern Territory are disproportionately disadvantaged, with inequalities not only limited to poverty, but also influenced by the broader social determinants of health, education, employment, skills development, technological innovation, transport and social support: " ... *it is imperative to reduce structural inequities in society through a more equitable distribution of community infrastructure resources, income, goods, and services for the holistic health and wellbeing of its people*". Residents, being the primary users of the community infrastructure, also identified the role that the extreme heat of Mparntwe plays in its utility. The community space of a Town Camp is necessarily outside and heavily reliant on structures that provide shade, trees and water. As the climate continues to get hotter and private power usage increases and becomes more costly, community centres are increasingly offering a place of last resort for the many Town Camp residents who suffer energy insecurity [18]. Facing some of the highest temperatures nationally, Town Camp communities are vulnerable to the effects of a warming climate. As Longden et al. point out, exposure to extreme temperatures is associated with a range of adverse health outcomes including death [18]. To provide some context, the 2004 report into Climate Change in the Northern Territory [19] noted that Alice Springs averaged 90 days over 35 °C and 17 days over 40 °C (in 2004). This report predicted that by 2030 these figures would increase to between 96–125 days over 35 °C and to between 21–43 days over 40 °C. The figures for 2018/19 have already surpassed these predictions [18].

## 3. Mapping Local Decision Making and Resilience

Participatory mapping has emerged as an important method to identify the values of a place [20]. Brown et al. found that the mapping of 'place values' includes land-use preferences and is generally stable over time, important characteristics of a robust *Making Local Decisions* methodology. Powell et al. similarly describe participatory mapping as being able to highlight and display the "*involuted relationships between self and place and the ways in which self and place are mutually constitutive and relational*" [21]. While mapping the phenomena shaping a person's individual experience of a place is a challenging concept [22], the LDM process is undertaken in discussion with the community as a group, with the resulting maps reflecting a "*collective rather than an individual outcome*" [23]. The *Making Local Decisions* methodology is outlined as follows:

- A high-resolution aerial photograph of the Town Camp is made from digital Nearmap images and printed out in large format, 1200 mm square. A small group of researchers, together with staff from Tangentyere Council, visit the Town Camp to initiate LDM discussions with the community. Meeting times often align with morning tea or lunch, with numbers and the make-up of the community group varying over the next hour or two;

- Depending on the availability of a community space, the aerial photograph is laid out on a large table in an inside or outside community space, allowing people to stand or sit around its edges (see Figure 1). The high-resolution of the image captures the smallest of details within the landscape, while also showing the broader organisation of the Town Camp and the roads and landscape that provide access to it. This is the only document brought in for the LDM discussion. None of the visiting group have clipboards, notebooks or any other equipment that differentiates them from the community;

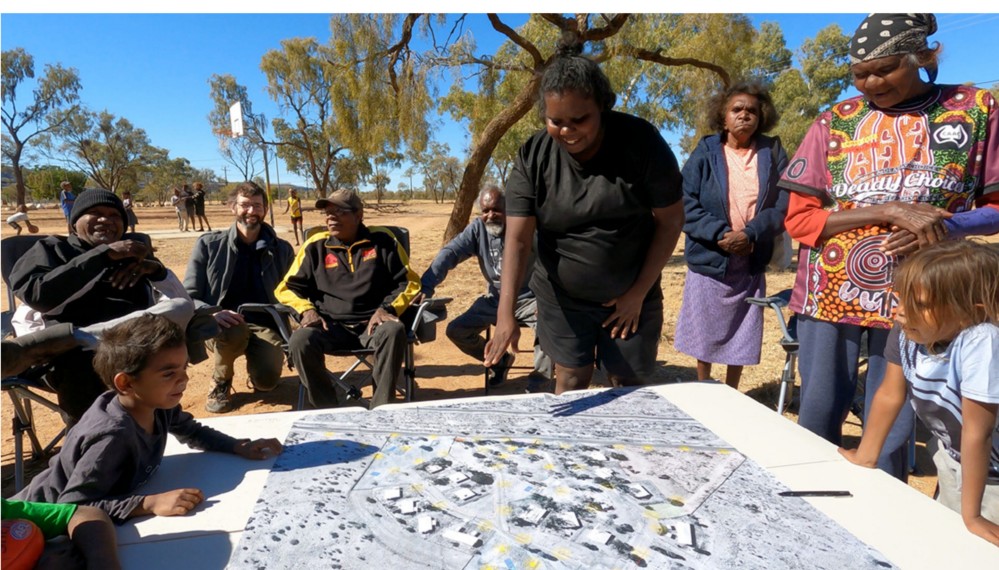

**Figure 1.** A Town Camp community gathering for a LDM Mapping process.

- The aerial photograph of the Town Camp is the centre-piece of the discussion, and people begin to engage with it immediately; its large size is novel but its content is relatable and easily understood. Fingers begin to run over pathways, and in a local language, residents point, discuss, laugh and gesture about what it shows. People find their own houses and they begin to discuss and tell stories about how the Town Camp works;

- The LDM process is introduced and then residents lead the conversation in a local language and sometimes in English. Issues with community infrastructure and housing weave in and out of conversations in different ways as resident groupings change over time. The accessibility of the aerial image invites contributions from all members of the community, no matter their age, literacy or language group. Children in particular appear drawn to the image; playful and enthused, they want to know what everything is while pointing out as much as they know;

- The community are encouraged to mark the image with felt-tip pens, locating: the routes of informal roads, occasional camping areas, broken street lights, breaks in fences, places that flood when it rains, bike tracks, routes people take when walking to town, or at night, the lack of playground fencing near fast moving cars, places for speed humps and pedestrian crossings, the lack of a road kerb and concrete pathways, and the lack of asphalt on roads (see Figure 2). These many discussions are about issues that other Australians and other residents of Alice Springs never have to live with. Discussions also reveal access to public transport, how people wait without shelter in extreme summer temperatures, and how energy insecurity poses significant health risks;

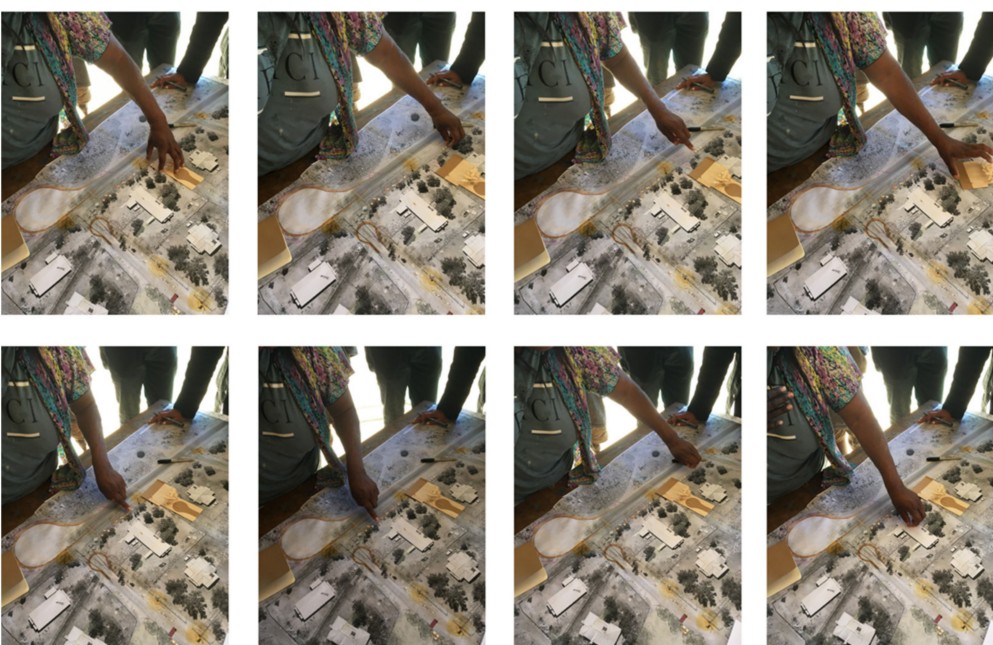

**Figure 2.** Examples of the LDM Mapping process from a Town Camp meeting.

- The aerial image becomes heavily marked in places with lines and symbols, disturbing its previously seamless qualities. We mobilise found objects from around the room and repurpose cardboard and timber blocks to represent buildings and structures. As the drawn marks and models grow in number, the aerial photograph gives way to the complexity of a topological map [24]. As the discussion highlights the past and future events of occupying the real space of the Town Camp, the visual qualities of the aerial photograph are transformed and differentiated to depict past events and issues and how the new projects will attempt to solve them. Each of these discrete projects is a topology, critically related and connected to others nearby. Now appearing on the map for all to see, apparent solutions to issues continue to be negotiated, edited and ultimately networked to each other as the mix of residents changes over time. *Mapping in this way is both a process and a tool for recording the conversations of Local Decision Making*;
- Following the meeting, the mark-ups left on the map are re-drawn digitally over the aerial image, with similar symbols being used as a record of the discussion. A few days later, the map with updated symbols is again printed out and a similar meeting is again made with the community to confirm, edit and add to what has been recorded. The map is again updated and refined to graphically depict Local Decisions as symbols. Models and more refined design drawings are also used where solutions will become community buildings or alterations to them (see Figures 3 and 4. A legend is now provided at the bottom of the map to confirm the meaning of the symbols to the community and for a broader audience that will follow (see Figure 5).
- When the Local Decisions within the map have been confirmed by the community, identified projects are tabulated in a schedule that both prioritises and itemises each element for costing. This schedule, together with the map, will form the basis of funding applications to territory and federal governments. As topologies, projects are also collated into the unpublished Guide to Infrastructure and Housing Standards for Town Camps (see Figure 6) so that similar issues and solutions between Town Camps can be identified and related to local, territory and national planning and regulatory requirements.

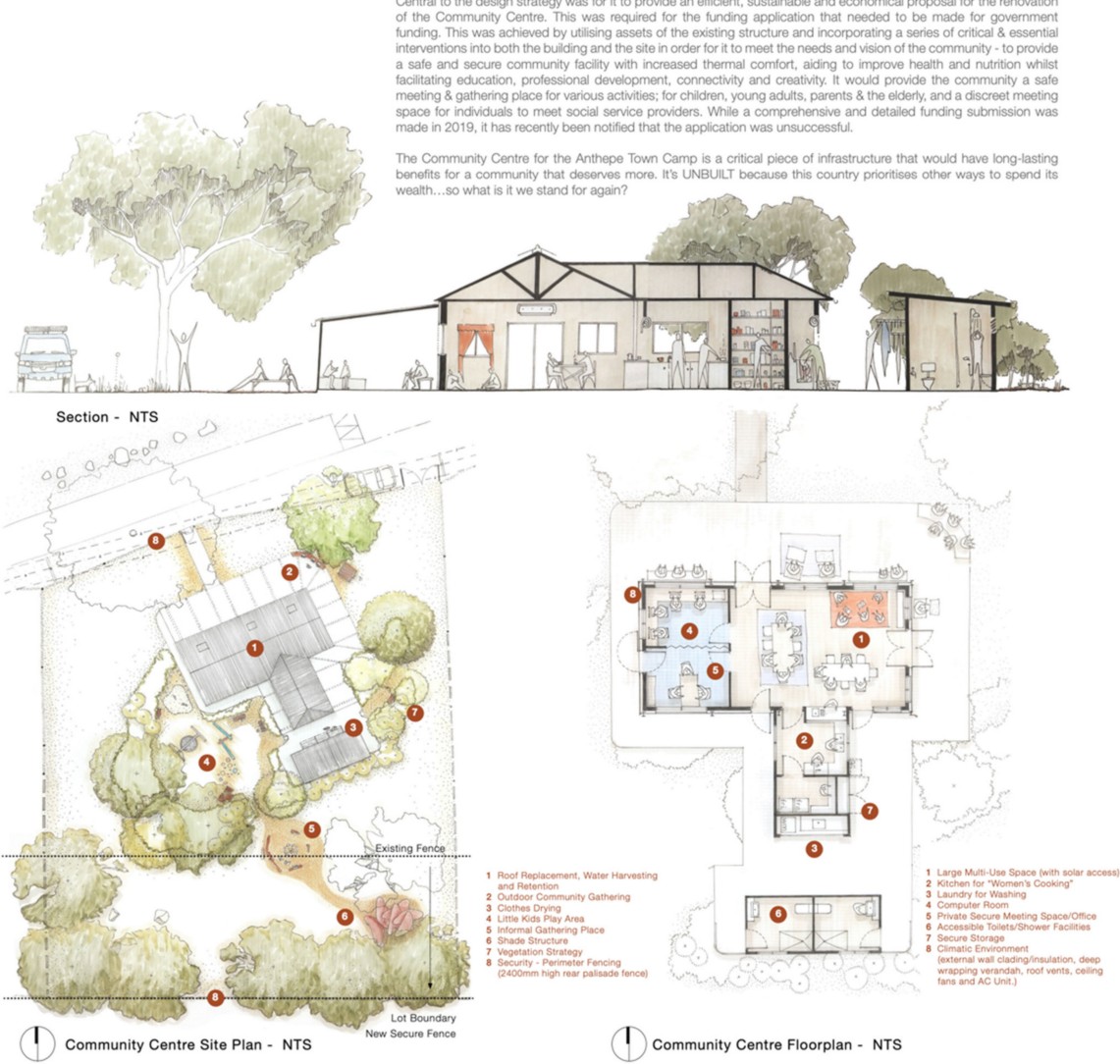

Central to the design strategy was for it to provide an efficient, sustainable and economical proposal for the renovation of the Community Centre. This was required for the funding application that needed to be made for government funding. This was achieved by utilising assets of the existing structure and incorporating a series of critical & essential interventions into both the building and the site in order for it to meet the needs and vision of the community - to provide a safe and secure community facility with increased thermal comfort, aiding to improve health and nutrition whilst facilitating education, professional development, connectivity and creativity. It would provide the community a safe meeting & gathering place for various activities; for children, young adults, parents & the elderly, and a discreet meeting space for individuals to meet social service providers. While a comprehensive and detailed funding submission was made in 2019, it has recently been notified that the application was unsuccessful.

The Community Centre for the Anthepe Town Camp is a critical piece of infrastructure that would have long-lasting benefits for a community that deserves more. It's UNBUILT because this country prioritises other ways to spend its wealth...so what is it we stand for again?

Section - NTS

Community Centre Site Plan - NTS

1 Roof Replacement, Water Harvesting
  and Retention
2 Outdoor Community Gathering
3 Clothes Drying
4 Little Kids Play Area
5 Informal Gathering Place
6 Shade Structure
7 Vegetation Strategy
8 Security - Perimeter Fencing
  (2400mm high rear palisade fence)

Existing Fence

Lot Boundary
New Secure Fence

Community Centre Floorplan - NTS

1 Large Multi-Use Space (with solar access)
2 Kitchen for "Women's Cooking"
3 Laundry for Washing
4 Computer Room
5 Private Secure Meeting Space/Office
6 Accessible Toilets/Shower Facilities
7 Secure Storage
8 Climatic Environment
  (external wall cladding/insulation, deep
  wrapping verandah, roof vents, ceiling
  fans and AC Unit.)

**Figure 3.** Proposed 2019 updates to the Anthepe Community Centre.

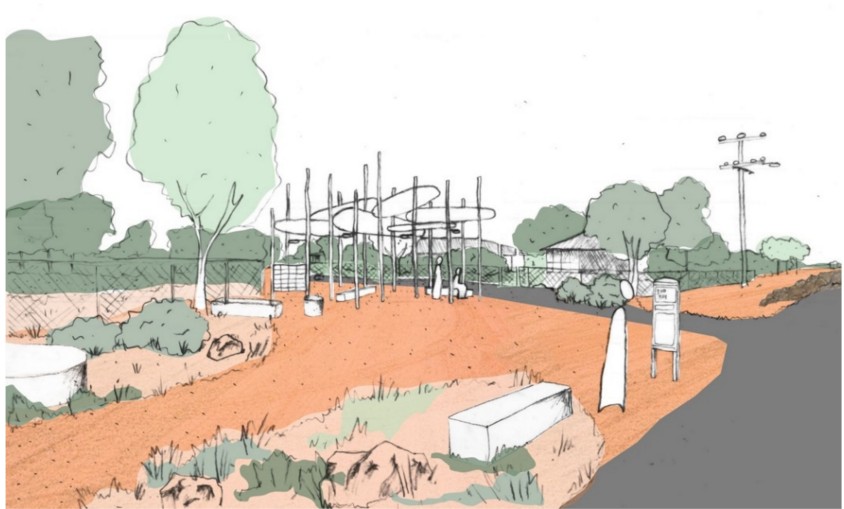

**Figure 4.** Sketch of shelter projects for Karnte.

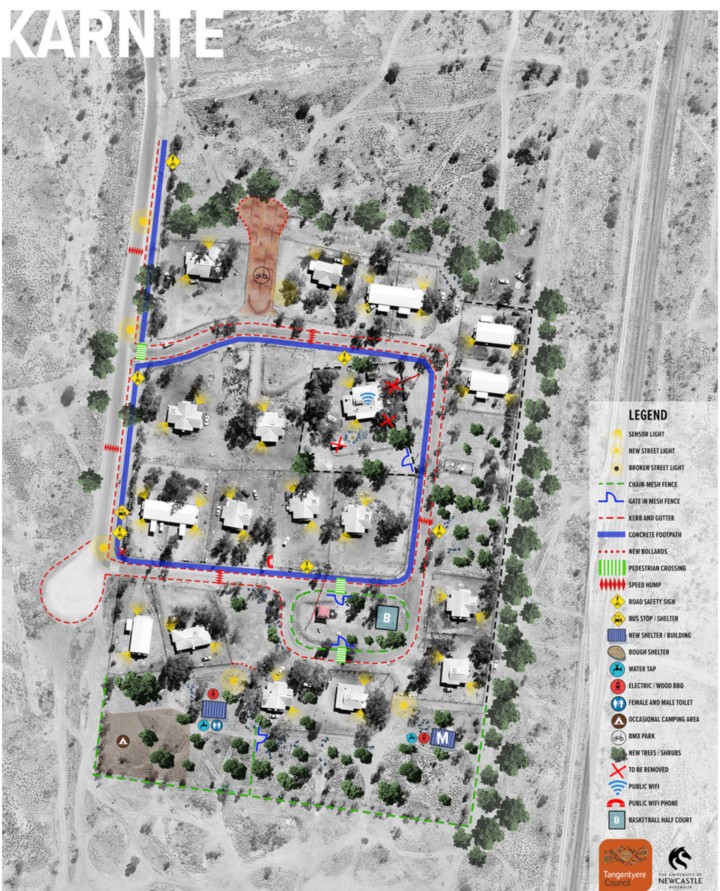

**Figure 5.** The LDM Map for Karnte.

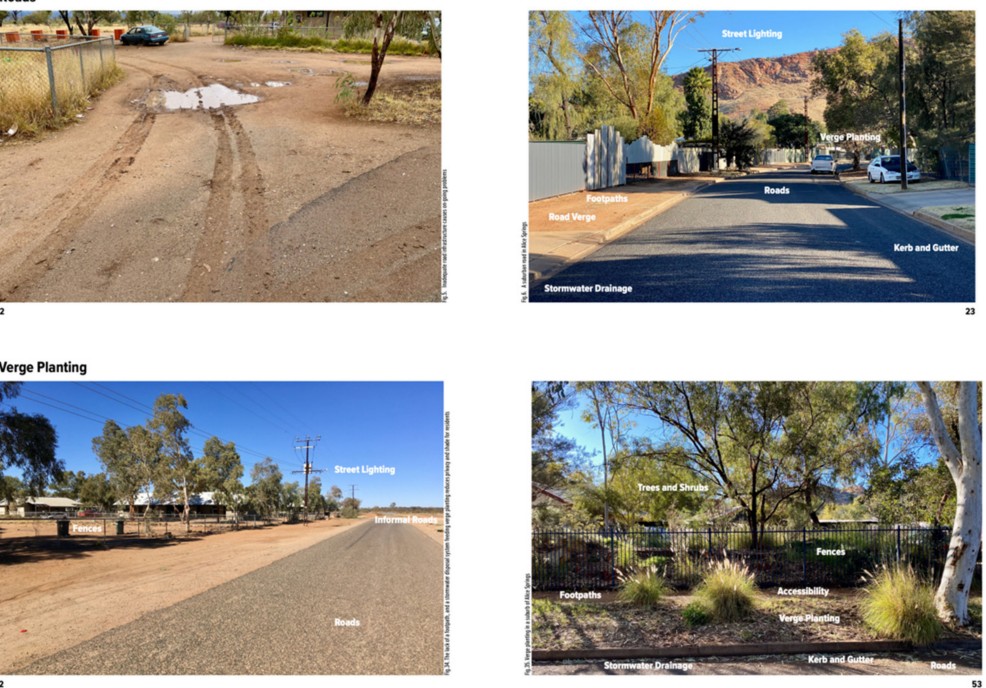

**Figure 6.** An extract from the Guide to Infrastructure and Housing Standards for Town Camps showing the 'Roads' and 'Verge Planting' topology. The situation found in the Town Camp on the left is contrasted with the same situation typically found outside Town Camps in Alice Springs.

## 4. Discussion

In the last few years of utilising *Mapping Local Decisions*, it has shown itself to be a highly engaging way to stimulate discussion about the future of the Town Camp. As Brown et al. reflect, "*In the transference of human knowledge and experience to a map through PPGIS, humans are reminded of their identity and dependence on place ...* " [23]. It invites a sharing of opinions that might otherwise not have been openly discussed and finds effective solutions to ongoing issues that may have never been found had a government-led *top-down* approach continued. The *Mapping Local Decisions* methodology, however, constantly evolves, with each Town Camp and each meeting providing greater insights into its process. The more time that is spent with the communities and engaged in *active listening* [25], the better we understand our role in translating local issues into architectural and community infrastructure proposals for government funding. The ongoing challenge of this research is to consciously work between two cultures [26], allowing conversations in the Town Camps to take their own shape, relying on the process of mapping to portray those discussions in accurate ways, and using more traditional architectural processes to communicate outcomes as funding proposals. The immediacy of working with the Town Camp community is vitally important to the research [27]; the non-verbal language [28], animation and position of those in the discussion is as important as anything that might actually be said. It allows relationships to be formed, sharpening an ability to apply architectural knowledge and develop skills that can quickly become useful.

In most towns and cities in Australia, communities don't need to engage with Local Decision Making and mapping processes to fund their community infrastructure. At some stage it was provided to them and remains essentially safe and well maintained; sealed roads have speed signs, stormwater drainage, kerbs and gutters, and there are footpaths, street lighting, pedestrian crossings and working community buildings. *The needs are so basic in Town Camps that it should embarrass a wealthy nation into immediate action; after so many years, why are we still talking about this?* The disparate state of community infrastructure within Town Camps reflects the colonial policies of inequity and dispossession, and the ever-changing funding landscape it has been subjected to [29]. As Senator Pat Dodson makes clear, the underlying issues of health, housing, education and employment need to be addressed as matters of urgency [30].

In terms of community infrastructure, the transition from suburban Mparntwe to the Town Camps is often stark. The usual asphalt roads become thinner, now with frayed edges as the kerb and gutter system disappears. With roads less defined, cars often leave the road entirely, creating a wider sandy zone that gets hollowed out and filled with water when it rains. This dusty edge optimises conditions for those that *drift* and travel at high speeds, making the side of the road a dangerous place for pedestrians. The ease with which cars can leave designated roads creates an informal road network that allows outsiders to enter the Town Camp in uncontrolled and dangerous ways. Those attempting to out-run the police also seek out the Town Camps, looking for these roads, knowing that their chase will be called off. At other times, police cars routinely prowl these informal roads for stolen cars and goods, keeping those who live nearby in a state of unease. Apart from the obvious solution of providing roads that meet minimum standards [31], with the required kerb and gutter [32], bollards can also be used to keep fast moving cars on the road and away from pedestrians. While road authorities and governments debate responsibility for these roads, none have the usual speed limit signs associated with suburban Mparntwe. More regular speed humps, designed to regulated standards, can also slow cars down. Stormwater needs to be collected at the side of the road as in nearby suburbs or placed in absorption trenches where it might help an already dry vegetated nature strip develop beside the road. Concrete pedestrian paths are almost universally non-existent in Town Camps and street lighting is a real problem, often well outside current regulation [33], located in ad-hoc and ineffective ways, and with bulbs not working. Where provided, community infrastructure is often so squeezed by the available funds that it is deployed in defensive, obligatory and loveless ways, detaching residents from its ownership and

the full potential of community space. As Crabtree et al. point out, the Town Camps of Mparntwe make the case that "*Aboriginal communities remain caught up in an ongoing melee of political opportunism, ideological posturing, dubious contractual dealings, and policy disjuncture, very little of which reflects or respects community experience, knowledge, or aspirations*" [34]. This, in many respects, is the importance of *Mapping Local Decisions*; it translates Local Decisions into community projects for government funding, making it clear within those documents how the existing infrastructure fails to meet the regulations and standards government has set for itself.

## 5. Conclusions

The PPGIS tools and methods utilised within *Mapping Local Decisions* has supported the effective inclusion of the community in LDM, with government funding beginning to flow into identified projects and some already constructed. These include new bollards in Ewyenper-Atwatyeare, constructed works at the A2E (Access to Education) Brown Street Youth Centre and Inarlenge Community Centre, upgrades to community and public infrastructure at Karnte, Anthepe, Anthelk-Ewlpaye, Lhenpe Artnwe and Ilyperenye, and accessibility upgrades to houses. Many other projects are documented and many of course are awaiting funding. While local government in Australia usually accounts for the inception and design of these types of community facilities, in the Town Camps it is organisations such as Tangentyere Council and the University of Newcastle that have come together to produce this work. It is in terms of this obligation that we have perhaps found better ways to engage across this cultural divide and fulfil a future that these resilient communities hold and have held for many decades.

Since 2019, when LDM was initiated, all 16 Town Camps in Mparntwe have undertaken some form of *Mapping Local Decisions*, with some continuing to refine and update proposals as circumstances change; residents see more potential in the process and appreciate the significant role it plays. The importance of housing and community infrastructure on public and environmental health outcomes cannot be overlooked. At present, the emphasis in the Northern Territory through the National Partnership Agreement is on new houses and new bedrooms. From the perspective of Tangentyere Council, the built environment and community infrastructure of the Town Camps and Remote Communities is a priority. These places need community infrastructure that meets Australian Standards and local government development guidelines, and recognizes the resilience and authority that local decisions made in Town Camps have in shaping the future.

**Author Contributions:** Conceptualization, C.T.; methodology, C.T., M.K. and A.F. All authors have read and agreed to the published version of the manuscript.

**Funding:** This research received no external funding.

**Institutional Review Board Statement:** Not applicable.

**Informed Consent Statement:** Informed consent was obtained from all subjects involved in the study.

**Data Availability Statement:** Not applicable.

**Conflicts of Interest:** The authors declare no conflict of interest.

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
