# Peer review of "Mapping Resilience in the Town Camps of Mparntwe"

_2673-8945, doi:10.3390/architecture2030025_

Round 1

Reviewer 1 Report

The abstract summarises well the paper.

This is a good paper on how to use PPGis for decisions. However, this where the shortcomings come from. The case study is abruptly introduced without properly referencing the methodology. There is a mailing list on PPGis for indigenous community, please consult for more references in order to be able to extract lessons for other sites. Particularly since the site is located in Australia, which is an industrialised country, the situation might differ from Africa. Also decision related literature is insufficiently referenced. PPGis for decision is not so spread as for mapping so given the innovation potential please comment more on its employment for example the relationship to games theory. There are games simulating resilince in the developing world for example. This will do also the necessary expansion to the references list.

The topic is relevant for the special issue considering SDGs such as well-being. 

The content of the paper itself: method, results and discussion is well done. The illustrations are good and necessary.

Reviewer 2 Report

The study examines the resilience among various communities in the concept of the Local Decision Making. The paper is interesting while I have some comments to improve the quality.

Abstract: This section should provide more concrete information of the study results and directions.

Intro: Introduction should start with resilience/equity/ and related more general scope and then delve into Alice Springs and other details.

Section 2-4 could be condensed and merged as they are very related to each other.

There is a great a section before Discussion to emphasize the findings/implications of the maps/resilience etc. Concepts of the study.

The study suffers from generalizbility concern even in Australia/

What is the main findings or outcome of the study? What is this community important to study? Any other agreements or ideas might be formed in a same way of the study? Why or why not?

Round 2

Reviewer 1 Report

The paper definitely improved compared to the earlier version. The references and the structure are now very good.

Reviewer 2 Report

Thank you for addressing my comments.